# Noninfectious Causes of Pregnancy Loss at the Late Embryonic/Early Fetal Stage in Dairy Cattle

**DOI:** 10.3390/ani13213390

**Published:** 2023-11-01

**Authors:** Zoltán Szelényi, Ottó Szenci, Szilárd Bodó, Levente Kovács

**Affiliations:** 1Department of Obstetrics and Farm Animal Clinic, University of Veterinary Medicine, 1078 Budapest, Hungary; szenci.otto@univet.hu; 2Institute of Animal Sciences, Hungarian University of Agriculture and Life Sciences, 2100 Gödöllő, Hungary; bodo.szilard@uni-mate.hu (S.B.); kovacs.levente@uni-mate.hu (L.K.)

**Keywords:** cattle, pregnancy loss, prediction, noninfectious, stress

## Abstract

**Simple Summary:**

Pregnancy loss at any developmental stage affects the reproductive efficiency of cattle both at the individual and herd levels. In dairy cows, negative effects on milk production along with the increased antibiotic usage associated with pregnancy loss can have dramatic consequences on herd economy. Pregnancy loss during late embryonic and early fetal development can affect up to 20% of animals within a dairy cattle herd, so regular screening and pregnancy maintenance protocols are mandatory as it is important to identify animals at risk of losing their pregnancy after its diagnosis. In this review, we describe possible noninfectious causes of pregnancy loss during this late embryonic/early fetal stage.

**Abstract:**

In cattle, initial pregnancy diagnosis takes place during the late embryonic/early fetal stage of gestation. From this point onward, pregnancy loss may occur in up to one fifth of pregnancies before the initial pregnancy diagnosis is confirmed. This means the early identification of risk factors is a key part of pregnancy diagnosis and herd management. The various factors responsible for pregnancy losses are classified into infectious and noninfectious. Among the noninfectious causes, several dam-related (circumstances of the individual pregnancy or milk production) and herd-related factors causing stress have been well established. In this review, we summarize the impacts of these noninfectious factors and predict associated risks of pregnancy loss.

## 1. Introduction

Dairy cattle show a higher incidence of pregnancy loss than other domesticated ruminant species, and these losses occur mostly at the late embryonic and early fetal stages of gestation. This timing affects the method of pregnancy diagnosis [1], such that after an initial diagnosis of pregnancy, its confirmation is required for acceptable farm-level clinical results. The embryonic developmental part of gestation ends on day 42 of pregnancy (Figure 1) [2]. From this point, fetal development ensues.

In a dairy cow, pregnancy can be first diagnosed only in the late embryonic stage of pregnancy approximately on gestation days 22–24. In practical terms, the recommended earliest pregnancy diagnosis time is on days 28–32 [3]. Hereafter, the general rate of losses under clinical circumstances ranges from 5% to 20% until day 60 of pregnancy [3], and thereafter, losses range from 1% to 5% [3]. Nevertheless, stillbirths can dramatically contribute to reproductive losses [4]. Risk factors for pregnancy losses are broadly classified into infectious and noninfectious. In this report, we review current knowledge regarding noninfectious risk factors as possible predictors of imminent pregnancy loss.

Because of the abovementioned risk of losing a successful pregnancy after a positive pregnancy diagnosis, there is a need to distinguish between dams showing a higher risk of late embryonic (days 30–42) and of early fetal (days 42–60) loss. In addition, it needs to be considered that more pregnancy losses occur in the late embryonic period. The factors responsible for pregnancy loss differ according to the developmental stage of gestation, late embryonic or early fetal. In clinical practice, there is no standard procedure available for the termination of these high-risk pregnancies as part of the management strategy. For instance, although Andreau-Vasquez et al. reported induced embryo reduction in twin pregnancies [5], this method has not become standard practice. So far, the genetic background of pregnancy loss has been addressed [6] and numerous predisposing factors have been identified [7]. In this review, we address the noninfectious factors described to date.

## 2. Noninfectious Causes of Late Embryonic/Early Fetal Losses

### 2.1. Dam-Related Risk Factors

#### 2.1.1. Declining Pregnancy Protein Concentrations

Reduced concentrations of placental lactogens in maternal blood have been identified as a marker of subsequent pregnancy loss [8,9]. However, in a clinical setting, the diagnostic potential of this marker is limited. Maternal blood concentrations of pregnancy proteins, arising from the fetoplacental unit, are elevated and measurable from days 22 to 25 of pregnancy [10]. Pregnancy-associated glycoproteins (PAGs) can be differentiated into two phylogenetic subgroups: PAG-2, mainly localized at the fetal–maternal border, and PAG-1, originating from trophoblast cells [11,12,13,14]. Currently, this distinction has no known consequence on pregnancy loss. In contrast, placental insufficiency is a known cause of late embryonic mortality. In this setting, cell protein production is disrupted [15]. Milk yield is a known animal factor that correlates with peripheral blood PAG-1 concentrations [16].

The accuracy of the measurement method should also be considered when performing a clinical diagnosis and analyzing data, as results will affect decision making at the time of the initial pregnancy diagnosis. Both early RIA [17] and ELISA tests [18,19,20,21] serve to accurately determine maternal blood concentrations of PAGs and pregnancy-specific protein B (PSPB). These tests are available in laboratories and as on-farm tests and can be used on biological fluids (whole blood, serum, plasma, and milk). Their overall sensitivity is 90% to 99%, whereas cow-side tests offer good results but of lower accuracy [22,23]. However, as these tests do not take into consideration the viability of the embryo [24], false positive diagnoses can occur in clinical practice. This determines that the prediction of pregnancy losses through pregnancy protein concentration measurements is difficult in practice. In effect, further research is warranted into pregnancy protein levels as predictors of pregnancy loss (Figure 2).

Fetal viability can be monitored through pregnancy proteins [24] as a drop in the level of these proteins occurs immediately after the death of the conceptus. PSPB and bovine PAGs have different half-lives [25,26]. Thus, a biological delay exists from the loss of the embryo/fetus and the decrease produced. In addition, differences have been detected between animals undergoing pregnancy loss and those not, and some studies have defined cut off values [15]. PSPB was recently monitored in more than 7000 early pregnancies [27]. The primary goal of this last study was to predict and identify factors detectable in cows carrying twins to determine the likelihood of twinning based on measured concentrations. Although plasma PSPB levels could not distinguish between cows carrying twins and singletons (2.1 vs. 2.9 ng/mL), mortality was higher in the case of singletons, yet the authors provided no explanation for the differences. In another recent study conducted in the US, plasma P4 concentrations differed in the case of unilateral twins, yet plasma PAG concentrations did not [28]. We also examined twin and singleton pregnancies in the first 4 months of gestation and, again, found no differences in PAG concentrations between twin and singleton pregnancies [29]. In contrast, Giordano et al. [10] were effectively able to confirm a difference in elevated pregnancy protein levels between twin and singleton pregnancies. These discrepancies highlight a need for repeated measurements to monitor risks of pregnancy loss in cows carrying twins, as the loss of one twin can occur in up to 10% of animals [30], and it is challenging to identify these animals. In these cases, transrectal ultrasonography (TRUS) and protein measurements are required. Clinical professionals currently propose weekly repeated tests in suspected cases of spontaneous embryo reduction [24].

**Figure 2 animals-13-03390-f002:**
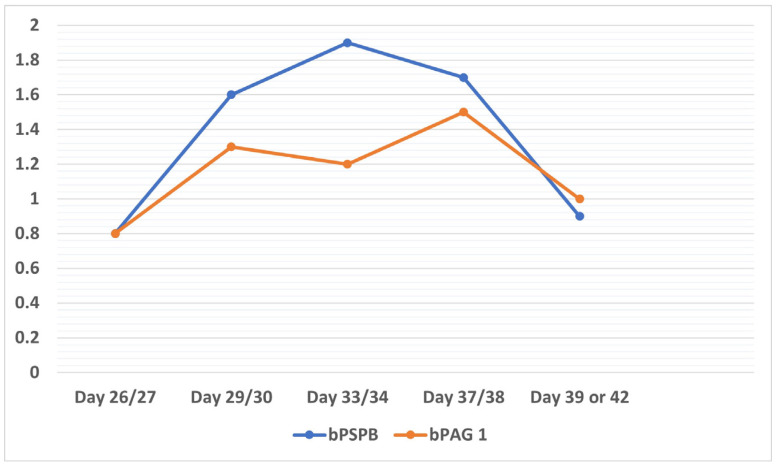
Decline produced in pregnancy protein concentrations measured in the same animal. Pregnancy loss was diagnosed via ultrasonography on day 42 of pregnancy. Adapted from Szenci et al. [26].

#### 2.1.2. Twin Pregnancy Diagnosis

An initial diagnosis of a twin pregnancy indicates an increased risk of pregnancy loss [4]. An incidence of twinning among all pregnancies in dairy animals as high as 20% has been reported [5]. However, in our setting, twinning rates at the time of early pregnancy diagnosis range from 5% to 12% [1], and pregnancy losses thereafter vary widely. According to recent reports, twin laterality can affect the pregnancy loss rate. Hence, a higher mortality rate has been detected for unilateral than bilateral twins (54.4% vs. 45.6%, *p* < 0.0001) [31], and this factor has been therefore defined as a clinically important noninfectious cause of pregnancy loss [31,32,33]. In bilateral twin pregnancies, when one embryo is located at the end of the uterine horn and the other occurs in the other horn but close to the bifurcation, the contents of the allantoic sac often extend into the contralateral uterine horn [33]. These cases should be assessed for loss risk as sometimes embryo reduction can be the solution to maintain the pregnancy [5]. In addition to twin laterality, milk production is also a key factor for pregnancy loss at the herd level that may affect pregnancy loss analysis. In unilateral twin pregnancies, treatment against pregnancy loss with GnRH was able to increase the rate of pregnancy survival [33]. These findings indicate that TRUS is needed both for the early pregnancy diagnosis of twins and subsequent exams in these animals to monitor losses.

#### 2.1.3. Number of Corpora Lutea

The number of corpora lutea present and their ultrasonographic structure are also important factors affecting pregnancy maintenance. Nonlactating non-pregnant dairy cows have smaller corpora lutea and higher peripheral blood progesterone concentrations than lactating animals [34,35]. Corpora lutea larger than 17 mm in diameter, as determined by TRUS, are considered mature [36]. In around 10% of all pregnancies, cows carrying singletons show an additional corpus luteum [27]. At the time of pregnancy diagnosis, 99% of singleton pregnant animals have one corpus luteum ipsilateral to the gravid uterine horn [4]. However, the transuterine migration of embryos has been confirmed and low numbers of contralateral corpora lutea detected in singleton pregnancies can be explained by migration of the oocyte [37]. However, a recent slaughterhouse study in *Bos indicus* cows was unable to confirm this [38]. We recently showed that animals with an additional corpus luteum did not undergo more pregnancy losses than those without an additional corpus luteum [4]. We propose that these animals already underwent partial pregnancy loss, possibly during the early embryonic development stage. This rationale was also proposed in another study, in which cows with an additional corpus luteum were found to be 8.3 times less likely to undergo pregnancy loss than those with no additional corpus luteum [39]. There have also been reports of positive correlation between corpus luteum blood flow and progesterone concentrations, and this may also indicate pregnancy maintenance [40,41,42]. In the late embryonic/ early fetal period, luteal regression at least 3 days after the death of the embryo has been described [43]. Doppler ultrasound scanning of uterine blood flow is also able to predict pregnancy loss [42,43,44].

The presence of a cavity in the corpus luteum can also be a cause of pregnancy loss. Although in non-pregnant animals the presence of cavitary corpora lutea has not been linked to lack of function, in that progesterone production and the response to prostaglandin treatments (luteolysis rate, ovulation rate) fails to differ from those observed in intact corpora [45,46], we observed a reduced pregnancy maintenance rate in animals with cavitary corpora lutea [4]. This observation highlights the need for strategies designed to elevate maternal plasma P4 concentrations, either by inducing an additional corpus luteum by medical treatment or by directly providing progesterone through an intravaginal device. These strategies have nevertheless yielded conflicting results [7,47]. The spontaneous occurrence of accessory corpora lutea provides a natural source of progesterone to maintain a pregnancy [7].

#### 2.1.4. Ultrasonographic Findings Indicating Imminent Pregnancy Loss

Through TRUS, it has been shown that the integrity and functions of embryonic/fetal particles and vesicles are crucial for conceptus viability. However, placentomes are also detectable during fetal development. From the onset of fetal development (day 43), both the amniotic and allantoic fluids are visible and distinguishable. Alterations such as reduced placental fluid volumes, increased amniotic fluid turbidity, and improper structure of the allantoic membranes are features observed during the TRUS examination of the ongoing process of pregnancy loss. Today, TRUS can be used as a real-time diagnostic method during the initial pregnancy examination [48]. When an embryonic/fetal heartbeat is lacking, a pregnancy loss diagnosis protocol should be implemented [24,31]. An embryonic/fetal heartbeat can also be detected by means of Doppler ultrasonography [42,49]. According to Kelley et al. [42], besides heartbeat monitoring, crown-rump length is also an indicator of imminent pregnancy loss. In their study, the mean crown-rump length of embryos carried to term was found to be greater on day 34 of gestation than in aborted cows. In more advanced stages of pregnancy (from days 40 to 95), other factors may affect the fetal heart rate [50]. One of these is possibly the onset of fetal movements. This explains why Doppler ultrasonography is essential for monitoring the later stages of gestation and why in clinical practice its use is not widespread for routine pregnancy diagnosis. It should be highlighted that for research purposes, Doppler ultrasonography provides more information than TRUS [51].

#### 2.1.5. Metabolic State at the Time of Pregnancy Diagnosis

Further clinically relevant factors associated with pregnancy loss are dam-related and include milk production, body condition score (BCS), and parity. A proper balance of carbohydrate intake is essential to maintain gestation and avoid pregnancy loss. While blood glucose levels are not adequate predictors of the fate of a pregnancy, some metabolic parameters such as excessive ketone body formation and increased blood urea nitrogen may suggest a threat to pregnancy. Further, several studies have shown that the pregnancy loss rate increases with number of parities or age [52,53,54]. Remarkably, higher pregnancy loss rates were observed in the third and fourth parities of Simmental cows compared to those of heifers, primiparous cows, and cows in their second lactation [55]. In contrast, Fernandez-Novo et al. [56] detected no effect of parity on conceptus mortality rates, reporting a 15% pregnancy loss rate in multiparous cows. However, losses as low as 3%–5% have been reported in primigravid animals [55,57,58]. In these studies, cows of parity three or higher always showed the higher rates of pregnancy loss, whereas its occurrence was lower in heifers and somewhat higher in primiparous cows. This means that parity distribution at the herd level is a useful reproductive management tool for predicting pregnancy losses [59]. However, although younger animals show lower pregnancy loss rates, a confirmation pregnancy diagnosis is still required in these animals [59]. Pregnancy loss impairs the subsequent reproductive performance of cows, causing serious economic losses, and mathematical models have been designed to estimate these losses [60]. Postpartum clinical diseases, such as mastitis [61,62], metritis [63], and metabolic problems [64,65], have been also identified as risk factors for pregnancy loss. Further, a carryover effect of postpartum inflammatory diseases has also been described [66]. Also considered as a dam-related predictor of pregnancy loss is energy balance. Carvalho et al. [58] recorded similar pregnancy loss rates in primiparous and multiparous cows with normal and low body condition scores. However, these authors noted poorer embryo quality in animals in worse conditions and multiparous animals. Further, a low BCS together with clinical mastitis was shown to impair reproductive function and increase pregnancy loss rates [62]. Additionally, a 10% rate of pregnancy loss was detected in animals with some clinical disease [67]. In contrast, no differences were found in pregnancy losses between cows that maintained their body condition and those with an impaired body condition in the first 30 days of lactation [68].

### 2.2. Environment-Related Factors

Early pregnancy can be compromised when cattle are subjected to environmental stress [69]. Individual animals experience different effects, in response to the same amount of stress. Stress affects uterine health, oocyte quality, ovarian function, and the developmental capacity of the conceptus [70]. Pregnancy loss in the early embryonic stage is more common than in the late embryonic or early fetal stages, and the loss of embryos younger than 16 days can be indicated by a return to estrus. This could be the reason for the relatively limited literature data regarding the role played by stress in early embryonic mortality in dairy cattle. Although stress can be accurately measured via physiological and behavioral indicators in dairy cattle [71], addressing this issue is challenging in the context of a multifactorial phenomenon such as pregnancy loss.

The numerous factors that lead to stress-induced pregnancy loss in individual animals can be classified in several ways. Nutritional stressors (quantitative and qualitative, animal welfare and climate effects both cold and warm) are classified as external stress factors while internal stressors include clinical disease, endocrine imbalance, and physical trauma (mechanical shock). The clinical impacts of stress on the reproductive system are mediated by some of the abovementioned factors. Among these factors, body temperature elevation (heat stress), metabolic hormones (stress related to mammary gland milk production and nutritional factors), and/or the activation of the hypothalamus–pituitary–adrenal (HPA) axis are important [72]. Activity of the HPA axis can be increased by several stressors. Examples of these stressors are new milking machines or housing conditions, negative interactions with herd mates, aversive human handling, or restraining. However, the effects of restraining, limited access to resources, and social stressors on late embryonic and early fetal development in dairy cattle are not as well understood. In a previous study, Szenci et al. [73] observed that restraining pregnant heifers for 2 h elicited a substantial increase in plasma cortisol concentrations 1 h after this restraint. However, this elevation of HPA axis activity did not affect late embryonic development between days 30 and 40 of pregnancy, as none of the heifers experienced embryonic mortalities. These results suggest that the termination of pregnancy is possibly the consequence of several staggered events. Limited space, elevated stocking density, and frequent regrouping can also cause social stress [74], such that younger cows may benefit from being housed in smaller groups with less competition [75]. Reduced dry matter intake has been associated with the increased expression of aggressive behavior [76] and this could negatively affect fetal development and survival through activation of the HPA axis. However, no clear effect of social stress on late embryonic/early fetal mortality has been shown. Kim et al. [63] reported increased late embryonic mortality rates in herds comprising fewer than 50 lactating cows versus more than 100 lactating cows. These results suggest that possible social stress related to greater herd sizes [77] is not necessarily associated with more late embryonic losses.

Environmental temperature has a clinically obvious effect on reproductive outcomes both in herds and individual animals. In a recent study by García-Ispierto et al. [78], a strong link was observed between pregnancy loss and heat stress. The likelihood of late embryonic mortality was increased by a factor of 1.05 for each additional unit of mean maximum temperature–humidity index increase recorded between days 21 and 30 of gestation. This increased index is probably also responsible for the loss of a unilateral twin pregnancy [32], which often occurs from days 60 to 90 [3]. The mean maximum temperature–humidity index has also been associated with higher ambient temperatures by other authors [31]. Late embryonic mortality varies widely in dairy cows in different temperature environments. After day 27 of gestation, this rate ranges from 3.2% under normal weather conditions [79] to 42.7% during periods of extreme heat stress [80]. This suggests that in the presence of heat stress, strategies against pregnancy losses should be initiated in clinical practice. Moreover, pregnancy should be monitored through combined pregnancy examination practices [59].

Transporting animals can also cause a high level of stress that could result in late embryonic/early fetal death. Based on the limited data available, it seems that the risk of mortality is higher in cows carrying later-stage embryos or early-stage fetuses compared to early embryos (less than 16 days of pregnancy). Yavas et al. [81] found that trucking stress for 1 h either before or after AI did not lower the conception rate among heifers. However, Harrington et al. [82] did record 9% higher pregnancy losses in cows transported from 29 to 33 days after AI compared to cows that were shipped within the first 4 days of pregnancy. Further, transportation for 4–6 h approximately 14 days after AI was found to induce acute stress without affecting pregnancy rates in beef cows [83]. These last authors proposed that the longer after gestation day 45, the less severe the influence of shipping stress on embryonic loss, as, at this time point, the embryo is well established and fully attached to the placenta [84]. Notwithstanding, other authors have reported 6% embryonic mortality in beef cows shipped 45–60 days after AI [85].

## 3. Conclusions

Several factors can be responsible in some measure for pregnancy losses in dairy cattle. In this review, we summarize individual factors (dam-related) and environment-related stress factors that may cause pregnancy loss. Several internal factors such as body condition or milk production serve to explain stress-mediated pregnancy loss in the individual animal. In the context of animal production, other effects may influence the success of pregnancy. Hence, the final clinical outcome can be predicted as the balanced consequence of these factors. This highlights the need for accurate diagnoses of pregnancy and of pregnancy loss. Pregnancy loss assessment commences in the initial pregnancy diagnosis exam and should consider all possible factors that could negatively affect the developing pregnancy and potentially result in its loss. The early identification of animals with a higher risk of losing their pregnancy offers economic benefits. This can be performed as early as in the initial pregnancy diagnosis. Avoiding negative consequences will also lead to improved herd profitability and reduced antibiotic usage. The different pregnancy determination methods currently available have different advantages in predicting pregnancy losses. Transrectal ultrasonography has the benefit of real-time observation of the viable embryo. However, it also has the drawback of requiring a skilled examiner. Certain visible changes in the placenta and the lack of an embryonic/fetal heartbeat are predictive signs of pregnancy loss that can be detected using this method. It should be mentioned that portable Doppler ultrasound devices designed for the cattle industry exist but are rarely used in clinical practice. For pregnancy monitoring, the measurement of pregnancy proteins is also an excellent approach, although the half-life of these proteins and the accuracy of the method used may limit their ability to successfully predict the termination of a pregnancy.

## Figures and Tables

**Figure 1 animals-13-03390-f001:**
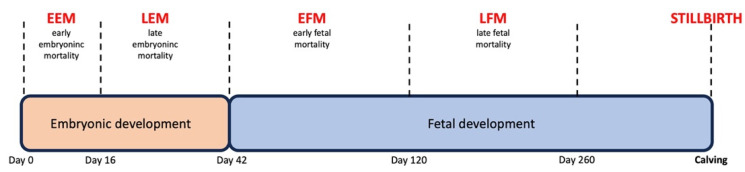
Developmental stages of bovine pregnancy and definitions of pregnancy losses.

## Data Availability

All datasets analyzed are found in public references.

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
