# Peer review of "Noninfectious Causes of Pregnancy Loss at the Late Embryonic/Early Fetal Stage in Dairy Cattle"

_animals, 2023, doi:10.3390/ani13213390_

Round 1

Reviewer 1 Report

Comments and Suggestions for Authors

Congratulations to the authors for this quite interesting review of late embryonic/early fetal mortality. The main non infectious factors have been correctly presented. 

I have some specific remarks or recommandations

Line 42 In his publication (Meta-analysis of the incidence of pregnancy losses in dairy cows at different stages to 90 days of gestation. JDS, 2023; 4:144–148),  Albaaj A et al.has well defined the different period namely early embryonic mortality (19 to 32 d), late embryonic mortality (30 to 45 d), early fetal mortality (45 to 60 d) and late fetal mortality (61 to 90 d). He has also proposed some prevalences according to 46 publications. It coud be usefull to present such definitions and associated prevalences.  A consensus is necessary. So such period can be associated with clinical method to confirm an embryonic/fetal mortality (P4, PAG, ultrasound B mode, echoDoppler).

Line 79 clinical practice

Line 92 and 96 repeated measurements : at which frequency is it necessary to take blood for PAG analysis ?

Line 103 could you mention the rates proposed by Lopez-Gatius ?

Line 105 what do you means by position of the two embryos

Line 122 This transuterine migration of embryos is very scarce

Line 134 See also the publication of Perez-Martin C. Formation of Corpora Lutea and Central Luteal Cavities and Their Relationship with Plasma Progesterone Levels and Other Metabolic Parameters in Dairy Cattle. Reprod. Dom. Anim., 2009,44,384-389 and Jaskowski BM et al. The impact of the cavitary corpus luteum on the blood progesterone concentration and pregnancy rate of embryo recipient heifers. Theriogenology 178 (2022) 73-76 : yhe % of pregnancy is higher in case of corpus luteum with cavity (47.7  vs 29.9 %).

Line 149 Kelley and not Kelly

Line 153 I don’t understand this sentence : This can be an explanation …

Line 175 : see also the publication of Pinedo et al. Association between body condition score fuctuations and pregnancy loss in Holstein cows. Journal of Animal Science, 2022, 100, 1–12 https://doi.org/10.1093/jas/skac266

Line 184 see alos these publications concerning the effect of heat stress on EM 

·         Santolaria P, López-Gatius F, García-Ispierto I, Bech-Sàbat G, Angulo E, Carretero T, SánchezNadal JA, Yániz J. 2010. Effects of cumulative stressful and acute variation episodes of farm climate conditions on late embryo/early fetal loss in high producing dairy cows. Int J Biometeorol, 54:93-98.

·         El-Tarabany MS and EL-Tarabany AA Impact of maternal heat stress at insemination on the subsequent reproductive performance of Holstein, Brown Swiss, and their crosses. Theriogenology 2015b; 84: 1523-1529. https://doi:10.1016/j.theriogenology.2015.07.040 .  

·         El-Tarabany, M. S., and K. M. El-Bayoumi. 2015. Reproductive performance of backcross Holstein× Brown Swiss and their Holstein contemporaries under subtropical environmental conditions. Theriogenology 83:444–448. https://doi.org/10.1016/j.theriogenology .2014.10.010.

·         García-Ispierto I, Lopez-Gatius F, Santolaria P, Yaniz JL, Nogareda C, Lopez-Bejar M. Factors affecting the fertility of high producing dairy herds in northeastern Spain. Theriogenology 2007;67:632e8

 Line 237 may bey ou ca add that the use of echoDoppler needs a quite lot experience.

Author Response

Thank you for your work and evaluation, here are the changes:

Please be aware that the other reviewers found too many (!) references so I try to add those ones which you find really necessary.

Line 42 In his publication (Meta-analysis of the incidence of pregnancy losses in dairy cows at different stages to 90 days of gestation. JDS, 2023; 4:144–148),  Albaaj A et al.has well defined the different period namely early embryonic mortality (19 to 32 d), late embryonic mortality (30 to 45 d), early fetal mortality (45 to 60 d) and late fetal mortality (61 to 90 d). He has also proposed some prevalences according to 46 publications. It coud be usefull to present such definitions and associated prevalences.  A consensus is necessary. So such period can be associated with clinical method to confirm an embryonic/fetal mortality (P4, PAG, ultrasound B mode, echoDoppler).

I think the difference of the nomenclature comes from the nature of the meta-analysis. The different authors are naming differently the gestational stages, we would like to stay at Day 42 as the border of embryoinic and fetal development based on the onset of organogenesis. If possible, I would like to leave in this publication because it’s a recent one with large number of analyzed data. Let me know if you have other opinion. (Finally I decided to prepare a Figure)

In line 42 Wiltbank and co-workers publication is cited whereas the third pivotal period ends at days 28. This falls into one with the first advised day of ultrasonography under practical circumstances some authors (Paul Fricke) advises Day 32.

Let me know the exact change in the wording if you request change.

Line 79 clinical practice

CHANGED.

Line 92 and 96 repeated measurements : at which frequency is it necessary to take blood for PAG analysis ?

Inserted the word: weekly repeated measurements 

Line 103 could you mention the rates proposed by Lopez-Gatius ? 

CHANGED

Line 105 what do you means by position of the two embryos

Inserted: the CLOSE position

Line 122 This transuterine migration of embryos is very scarce

In an earlier publication we used this term, but now I found a reference from a slaughterhouse study which says the opponent. This reference is also added in the sentence.

Line 134 See also the publication of Perez-Martin C. Formation of Corpora Lutea and Central Luteal Cavities and Their Relationship with Plasma Progesterone Levels and Other Metabolic Parameters in Dairy Cattle. Reprod. Dom. Anim., 2009,44,384-389 and Jaskowski BM et al. The impact of the cavitary corpus luteum on the blood progesterone concentration and pregnancy rate of embryo recipient heifers. Theriogenology 178 (2022) 73-76 : yhe % of pregnancy is higher in case of corpus luteum with cavity (47.7  vs 29.9 %).

One of our findings was in the cited study, that cavity presence did increase pregnancy loss, this is stated in the line. Because Jaskowski et al is a recent literature, but made in heifers and diagnostic imaging was carried out ont he day of ET and pregnancy was checked by manual palpation, I would not change the sentence, if possible. However, thank you for observing this literature.

Line 149 Kelley and not Kelly

CHANGED

Line 153 I don’t understand this sentence : This can be an explanation …

CHANGED to : The phenomenon can be an explanation...

Line 175 : see also the publication of Pinedo et al.

I re-read the sentence: probably controversial studies can be found in the literature. The cited study demonstrates in different phases of the lactation difference in the pregnancy loss with relatively low odds ratio. Let me know if you request a change.

Line 184 see alos these publications concerning the effect of heat stress on EM  

HAVE CHECKED THEM. As I said upon, a criticism of the manuscript was the large number of references.

 Line 237 may bey ou ca add that the use of echoDoppler needs a quite lot experience.

A HALF SENTENCE ADDED.

Reviewer 2 Report

Comments and Suggestions for Authors

Authors summarized some of the noninfectious causes of pregnancy loss in cattle. 

In my humble opinion, this mini-review summarized some of the main noninfectious effects of pregnancy loss in cattle. However, they did not deep in each one, comparing results, hypothesis and highlighting the main points of each posible pregnancy loss factor/cause. I think they just describe some outputs from different studies throughout the manuscript, including a good number of references, but, it has serious problems about language flow, and also, I think it is not strong enough as a revision. 

I think authors have to re-think this review and keep working on it prior its publication.

Some other comments:

- ln 134. You stated that is an important factor, but then you do not describe that importance...

- English revision is recommended

- Format is not as guidelines

Comments on the Quality of English Language

Extensive editing of English language required

Author Response

Thank you for your work and your opinion. First of all, english review has been performed, we hope it has improved the manuscript. Formatting has also changed according to the guidelines, in this matter we are in continuous connect with the edit. According to other reviewers requests:

-summary and abstract has changed but we have to keep the character limitation

-other reviewers requsted not to increase the number of references. Therefore some parts are not only reworded than rewritten, we try not to involve more references.

- regarding to the content: the goal of the review was to summarize to noninfectious causes of losses from the viewpoint of the clinican. In case of herd level increased pregnancy losses (which occur on a daily basis in the reproductive work) a diagnostic algorhytm must be evaluated, our aim was to give aid for those ones who are doing this on a daily basis.

Reviewer 3 Report

Comments and Suggestions for Authors

Since it is not a research article, I believe that a more careful and detailed description of the described phenomena is necessary. In the attachment you will find the manuscript with comments inserted and highlighted in yellow that needs to be detailed.

Line 37: why at 42 days, it is necessary to strengthen the affirmation, detail the physiological phenomenon.

Lines 43-44. This phrase needs citation, please specify a reference where these values can be found.

L 45-46 rephrase, I don't understand what you want to say.

Premature birth with a dead fetus is the same as abortion, pregnancy losses are just that - abortion.

L 51: Define the terms and gestational interval. Late embryonic and early fetal loses.

L: 53-54: Please rephrase, I don't understand what you mean

2.1.1.

  these proteins should be written more prominently.

detailed and in parentheses the abbreviation.

It is good to describe the technique for each, the period, the specificity

L: 100. Confused phrase! I disagree.

How does the diagnosis of twin pregnancy increase the risk of pregnancy loss??

L 134: CL cavitary - does not affect the pregnancy loss rate !!!

2.1.2. More information should be provided about:

- Accessory CL, what is how it develops, spontaneous and induced. Likewise, his relationship with the P4 level.

Eco Doppler evaluation of gestational CL and the relationship with P4 and pregnancy loss

L 143 Information is needed on the definition of the Fetus and Embryo phases. Try to insert the definition, period of development, and diagnostic methods. How is it differentiated sonographically?

2.1.4.

The metabolic profile should be detailed here, as the title shows.

you have to detail about energy, glucose, urea, micro and macro elements. How does their level (deficiency or excess) influence embryonic death / fetal development!

It also details (in addition to parity), the relationship between the level of milk production (possibly PRL) and pregnancy loss.

You also introduced some examples of pathology, but without detailing them... so it should also be reformulated with the biochemical changes in case of diseases and how pregnancy is lost.

Don't forget the relationship between nutrition and pregnancy loss!, mycotoxins... is very important in a dairy cow industry.

2.2.

This subchapter needs to be rewritten, I recommend breaking it down by category:

-Nutrition (feed-rate) quantity and quality + toxins

-Welfare (compare accommodation systems, BCS, ..)

-Climatic factors (thermal stress - counteracting conditions)

-Internal stressors (diseases) detail the effect of corticoids, hormones, adrenaline, etc. on pregnancy loss.

-trauma, mechanical shocks (pfysical causes)

-and others. (breed, calving-conception intervals, vaccination). depending on your ability.

The article still has a long way to go before it can be accepted for publication. But with the required improvements, it can get there.

Author Response

Reviewer 3

Thank you for your extensive review, here are my answer tot he requested changes

Line 37: why at 42 days, it is necessary to strengthen the affirmation, detail the physiological phenomenon.

ADDED.

Lines 43-44. This phrase needs citation, please specify a reference where these values can be found.

If I put in the previous one (Wiltbank et al, 2016) that contains descriptive data. Numerous citation is existing, if you request another, please let me know.

L 45-46 rephrase, I don't understand what you want to say.

Premature birth with a dead fetus is the same as abortion, pregnancy losses are just that - abortion.

Thanks for the correction, I rephrased.

L 51: Define the terms and gestational interval. Late embryonic and early fetal loses.

MADE THE CHANGE

L: 53-54: Please rephrase, I don't understand what you mean

REPHRASED

2.1.1.

  these proteins should be written more prominently.

detailed and in parentheses the abbreviation.

It is good to describe the technique for each, the period, the specificity

Did some change. The chief editor also requested some words, so upon your request this can be further modified.

L: 100. Confused phrase! I disagree.

How does the diagnosis of twin pregnancy increase the risk of pregnancy loss??

The meaning of the sentence: when performing pregnancy test we both diagnose singletons and twins. Simply the twin diagnosis highlists that higher risk of PL can exist compared to singleton one. I reworded the sentence with one single word.

L 134: CL cavitary - does not affect the pregnancy loss rate !!!

As the sentence says: we observed once differently, and was once again observed in an unpublished dataset. If possible I would like to leave this half sentence in.

2.1.2. More information should be provided about:

- Accessory CL, what is how it develops, spontaneous and induced. Likewise, his relationship with the P4 level.

This information is mentioned in the last 3 sentence of the chapter. Minor addition I made, if you request more let me know.

L 143 Information is needed on the definition of the Fetus and Embryo phases. Try to insert the definition, period of development, and diagnostic methods. How is it differentiated sonographically?

Inserted one half-sentence to the end of first sentence: Whereas in the fetal development placentomes are also observable. Is it enough?

2.1.4.

The metabolic profile should be detailed here, as the title shows.

you have to detail about energy, glucose, urea, micro and macro elements. How does their level (deficiency or excess) influence embryonic death / fetal development!

It also details (in addition to parity), the relationship between the level of milk production (possibly PRL) and pregnancy loss.

You also introduced some examples of pathology, but without detailing them... so it should also be reformulated with the biochemical changes in case of diseases and how pregnancy is lost.

Don't forget the relationship between nutrition and pregnancy loss!, mycotoxins... is very important in a dairy cow industry.

In this part we mainly focused on tools/practices that are under practical circumstances as first aids can help the decision making for the professionals. This chapter/part could be issue for another review, but here -according to our explanation- it is not underlined that much. Despite that, changes (extensions and rewording) have been made. Let us know if you request more.

2.2.

This subchapter needs to be rewritten, I recommend breaking it down by category:

-Nutrition (feed-rate) quantity and quality + toxins

-Welfare (compare accommodation systems, BCS, ..)

-Climatic factors (thermal stress - counteracting conditions)

-Internal stressors (diseases) detail the effect of corticoids, hormones, adrenaline, etc. on pregnancy loss.

-trauma, mechanical shocks (pfysical causes)

-and others. (breed, calving-conception intervals, vaccination). depending on your ability.

The article still has a long way to go before it can be accepted for publication. But with the required improvements, it can get there.

All your requested changes are written into this subchapter and as a result of this the final conclusions part is also becoming longer.

Reviewer 4 Report

Comments and Suggestions for Authors

This is a review on the causes of late embryonic and early fetal mortality. The article reviews in detail the causes related to the mother and the environment and includes a part on the early diagnosis of these pregnancy losses.

The review is well prepared, from the point of view of novelty, the most recent bibliography included deals with diagnosis and environmental causes.

From the point of view of the structure, causes and diagnostic methods are mixed in section 2.1. I suggest making a separate section for diagnosis that includes pregnancy proteins and ultrasound.

Lin 120: Delete the written quote and add the number.

Line 148: Delete the square brackets.

Author Response

Thank you for your work and effort. The two requested changes regarding to references have been done.

Regarding the structure of the sections: the title of 2.1 section is dam-related causes of pregnancy losses. This review wants to focus to these from the viewpoint of the clinical veterinarian therefore there was a debate between authors about the arrangement of the sections. Since the main goal of the review is not to summarize diagnostic possibilities than setting up a “checklist” for clinicians to run over in case of increased ratio of pregnancy losses, after discussion of your request we kindly ask to leave in its current form. Moreover, other reviewers have also suggested some modifications in the points, that have been performed. Please let us know if you cannot pass the revision without these.

Reviewer 5 Report

Comments and Suggestions for Authors

The review article is a summary of existing knowledge on the effects of various factors on embryo death in late embryo/early fetal in cows.
Overall the manuscript is written very well beyond Simple Summary and Abstract. These sections need to be completely rearranged because they do not relate to the content.
Typos should be checked throughout the manuscript.
There are far too many references that are already outdated, I recommend limiting them to the most recent items.

Author Response

Thank you for your work and observations. After incorporating your and other reviewers’ recommendations we did noticeable changes on the manuscript. Only you (and the editor) requested changes in the summary and the abstract please read over the changes: the main goal was to make them in accordance with the manuscript.

Regarding to the references: it is not easy, other reviewers requested further references. We tried everywhere to limit their number as low as we can. The typhos have been checked, and we have done a language revision as well.

Round 2

Reviewer 1 Report

Comments and Suggestions for Authors

I have still some remarks 

In Figure1 : embryonic and not embryoninc

Line 39 :  I believed like written by Caton et al. (2020 : doi:10.1093/jas/skaa358 ) that by approximately day 45 of gestation in cattle, organogenesis (development of the fetal organs) is completed for some organs; however, although differentiated, most organ systems like the gastrointestinal system (Noah et  al., 2011) or lungs are not completely developed or fully functional : so the organogenesis don’t begin at day 42. .

Line 135 : such migration is relatively scarced

Line 147 You mention « A cavity in the corpus luteum is also essential to pregnancy loss » Could you tell more precisely if this cavity is or not a risk factor of embryonic mortality. See Jaskowski BM et al. The impact of the cavitary corpus luteum on the blood progesterone concentration and pregnancy rate of embryo recipient heifers. Theriogenology 178 (2022) 73-76.

Line 156 :  2.1.3. Ultrasonographic findings of the imminent pregnancy loss : this paragraph is not really a cause of embryonnic/fœtal mortality.

Line 274 The technical circumstances  are rare now : why such circumstances are rare ?

Author Response

Firstly, thank you very much again, for the extensive review and your work.

Figure1 : embryonic and not embryoninc

Corrigated, thanks

Line 39 :  I believed like written by Caton et al. (2020 : doi:10.1093/jas/skaa358 ) that by approximately day 45 of gestation in cattle, organogenesis (development of the fetal organs) is completed for some organs; however, although differentiated, most organ systems like the gastrointestinal system (Noah et  al., 2011) or lungs are not completely developed or fully functional : so the organogenesis don’t begin at day 42. 

As I ran over the publication, finally the authors suggest Day 50 as the endpoint of embryonic development, furthermore it seems for me in the manuscript, that they will soon suggest that different housing and feeding environment influences the onset of organogenesis and the process of embryonic development. I suggested therefore to leave out the hall half sentence - anyhow, thanks for the correction about the organogenesis.

Line 135 : such migration is relatively scarced

You mentioned it in the previous 2 review as well. We discussed again. Because in previous 2 articles this term was not a problem we kindly ask to leave it in and in the future we will try to explain with other words the causes of the small amount contralateral pregnancies in case of singletons.

Line 147 You mention « A cavity in the corpus luteum is also essential to pregnancy loss » Could you tell more precisely if this cavity is or not a risk factor of embryonic mortality. See Jaskowski BM et al. The impact of the cavitary corpus luteum on the blood progesterone concentration and pregnancy rate of embryo recipient heifers. Theriogenology 178 (2022) 73-76.

Sorry, I don't even know how this sentence could be left in the manuscript.  I changed the sentence: A cavity in the corpus luteum can be source of pregnancy loss. Thanks for the observation.

Line 156 :  2.1.3. Ultrasonographic findings of the imminent pregnancy loss : this paragraph is not really a cause of embryonnic/fœtal mortality.

That is true, but somewhere earlier I wrote about the theoretical structure of the review (maybe to another reviewer): the idea was to prepare a decision-tree like overview the clinicians/professionals. In this manner, not the detailing of the diagnostic techniques, just the overview of the crucial points is in our view essential: during setting up the initial clinical pregnancy diagnosis without taking samples only the ultrasonography is the tool to predict pregnancy losses. The fact is obvious, but we would like to leave in the key diagnostic points if possible.

Line 274 The technical circumstances  are rare now : why such circumstances are rare ?

modified the sentence. At the moment the portable ultrasounds are not carrying Doppler mode, but IMV company is right now in this summer comes out with that possibility: this will change our sentence soon.

Again, thank you for your cooperation.

Reviewer 2 Report

Comments and Suggestions for Authors

As I said in my previous revision: 

In my humble opinion, this mini-review summarized some of the main noninfectious effects of pregnancy loss in cattle. However, they did not deep in each one, comparing results, hypothesis and highlighting the main points of each posible pregnancy loss factor/cause. I think they just describe some outputs from different studies throughout the manuscript, including a good number of references. I think it is not strong enough as a revision. 

I think authors have to re-think this review and keep working on it prior its publication.

Comments on the Quality of English Language

Minor editing

Author Response

Thank you again for your review. We tried to summarize your and other author's intstructions and tried to build them in into the manuscript. We will wait the other round 2 reviews and will try to make further efforts. Other reviewers named specific lines for detailing and reconstruction of the manuscript, if you see the manuscript in reviewer mode you can see that many changes have been done including those, which required deeper description. English has also improved,if the editor requires further improvement we're open to use the english grammar service of the company.

Reviewer 3 Report

Comments and Suggestions for Authors

I carefully read the manuscript revised by the authors.

It seems that the authors are on the right track.

The author brought arguments to most of the comments, but in some comments (especially the last two that referred to List R1 at point 2.1.4 and 2.2.) did not bring the expected changes.

Author Response

Thank you for your comments, we also feel that the reviewing has improved the quality of the manuscript. The parts that you mention, are primarily written by dr. Kovacs whos has significant publications in the field of stress. After discussing with him, we changed and built in your observations. We think now that this part deserves a seldom review in longer extension and in this review we would like to leave in this checklist-like version for the practitioner professionals. Regarding to stress-causing environmental effects associated with pregnancy loss to literature is surprisingly poor, so we also think in clinical studies as well. Again, thank you for your review!

We corrigated english language as well. As not a native speaker we have nothing left, that if the editor's decision is english proofreading we will take the publisher's service.